# Polyply; a python suite for facilitating simulations of macromolecules and nanomaterials

Fabian Grünewald [1], Riccardo Alessandri [1,2], Peter C. Kroon [1], Luca Monticelli [3], Paulo C. T. Souza [3] & Siewert J. Marrink [1 ✉]

Molecular dynamics simulations play an increasingly important role in the rational design of (nano)-materials and in the study of biomacromolecules. However, generating input files and realistic starting coordinates for these simulations is a major bottleneck, especially for high throughput protocols and for complex multi-component systems. To eliminate this bottleneck, we present the polyply software suite that provides 1) a multi-scale graph matching algorithm designed to generate parameters quickly and for arbitrarily complex polymeric topologies, and 2) a generic multi-scale random walk protocol capable of setting up complex systems efficiently and independent of the target force-field or model resolution. We benchmark quality and performance of the approach by creating realistic coordinates for polymer melt simulations, single-stranded as well as circular single-stranded DNA. We further demonstrate the power of our approach by setting up a microphase-separated block copolymer system, and by generating a liquid-liquid phase separated system inside a lipid vesicle.

---

[1] Groningen Biomolecular Sciences and Biotechnology Institute and Zernike Institute for Advanced Materials, University of Groningen, Groningen, The Netherlands. [2] Pritzker School of Molecular Engineering, University of Chicago, Chicago, IL 60637, USA. [3] Molecular Microbiology and Structural Biochemistry, UMR 5086 CNRS and University of Lyon, Lyon, France. ✉email: s.j.marrink@rug.nl

Molecular dynamics (MD) simulations of (bio-)macromolecules have become a powerful tool for researchers to complement experimental assays. Whereas simulations of single polymer melts or mixtures have been used since the advent of modern MD[1,2], recently the trend goes towards studying more complex multicomponent systems either of purely synthetic materials or biologically synthetic hybrid macromolecules[3–9]. Examples of such systems range from polyelectrolyte complex coacervates[10] to next-generation polymer batteries[11,12]. Whereas simulations of these complex systems are typically focused on studying the self-assembly or understanding structure–function relationships, much effort is now being directed towards developing MD-based protocols for virtual high throughput (HT) screening of polymers as exemplified by the material genome initiative[13–16]. HT screening of polymers by MD is expected to complement experimental HT approaches because it is typically less costly than synthetic exploration and gives access to properties not easily accessible by purely experimental HT approaches. Such combined approaches will enable researchers to survey a larger combinatorial space and filter possible candidates more efficiently[14,16]. Applications of such procedures range from the design of novel antimicrobial polymers to biodegradable polymers[14].

Though the avenue of HT simulations is promising, it requires programs to build topologies and simulation boxes in a quick, reliable, and consistent manner. Moreover, given the hierarchy of spatiotemporal scales underlying the behavior of polymer-based systems, models with both all-atom and coarse-grained (CG) resolution are required. While a wide range of programs[17–26] is available for MD simulations of biologically relevant systems such as proteins, lipid membranes, and DNA, the support for simulation of synthetic and biosynthetic hybrid macromolecules is largely lacking. To our knowledge, there are no programs that can generate both input parameters and coordinates for arbitrarily complex polymeric systems independent of the force-field and compatible with HT approaches. Depending on the molecule or system there are a number of specific solutions[20,27–30]. Some of those are in principle capable of generating parameters and coordinates[29,30]. However, available programs typically support only one force field and are limited to specific (mostly linear) polymers implemented by the developers. Website implementations[28,30] have the added problem that performance relies on server-load and it involves human-time having to interface with the website. In addition, coordinates for more complex systems such as micro phase-separated polymers and hybrid nanoparticle blends are frequently generated by (multi-scale) self-assembly[31–33] or custom in-house building scripts[34,35].

The general lack of programs supporting all-atom and CG polymer simulations limits the use of MD simulations for both large versatile systems and HT research of (bio-) macromolecular systems. This is especially true for non-experts in the field. To this end, we identify five major challenges that need to be overcome. (1) The program needs to be able to generate both coordinates and parameters, resolution and force-field independent. Accurate CG models are often based on atomistic polymers, thus modeling both of those is an integral part of HT model development. In addition, the program needs to be force-field agnostic, as some force-fields work better for specific polymers than others. (2) There needs to be an easy-to-use pipeline for generating input files and coordinates based on the system composition. Input files should be generated from sequences of arbitrarily complex polymers, including different degrees of branching and statistical distribution of residues along the chain. (3) The program needs to be able to combine input parameters and coordinates of polymeric systems with a variety of biomolecular structures, like proteins, lipid bilayers, and nucleotides. For example, manipulation of proteins and other biomolecules by polymer grafting is an integral part of enhancement strategies[36]. (4) It needs to be capable of setting up complex systems without the need for extensive relaxation. Polymer melts, blends with nano-particles, and phase-separated systems are highly important in material science, and when studying bio-synthetic hybrid molecules, and closely capturing their heterogeneity in the starting structures saves valuable computer resources. (5) Generation of both the coordinate and parameter files needs to be fast enough to enable HT research.

Here, we present the open-source polyply software suite which addresses the five major challenges presented above. It facilitates the generation of input parameters and coordinates for MD simulations of (bio-)macromolecules and nanomaterials. Using a graph-based algorithm, polyply allows users to generate parameter files of arbitrarily composed and branched polymers for any force-field from simple library files and the residue graph. A residue graph contains the sequence of residues of the polymer, but in addition, it also records which residues are connected. Using a multiscale random walk, polyply can also be utilized to generate starting coordinates for any force field and at any target resolution. This includes complex arrangements like microphase-separated polymeric systems or multi-component polymer solutions enclosed in lipid vesicles. To maximize the accessibility of the models and code, polyply is distributed via the python package index. Furthermore, polyply is developed using modern software development practices (such as code review, continuous integration testing, and semantic versioning) as outlined in the recent whitepaper by the BioExcel consortium[37]. These practices ensure both the integrity of the code and the library data files.

The remainder of this paper is organized as follows. First, we present the algorithms behind the input parameter generation and the coordinate generation. Subsequently, we show the capabilities of our program based on three examples from pure materials science to bio-molecular science, some at the atomistic level based on the Amber[38,39] and GROMOS[40] force fields, others at the CG level using the Martini[41,42] force-field. They exemplify the capabilities of polyply to be compatible with HT approaches, generate parameters force-field independently and set up complex systems. Finally, we discuss the limitations of our approach and sketch possible future directions for its further development.

## Results

The general code design accounts for the fact that generation of input parameter files and coordinate generation are in principle separate problems. However, both problems can make use of the same infrastructure, which is centered around exploiting graph representations of molecules. Thus, the polyply software suite consists of separate lone-standing programs, which utilize the same libraries. At the moment two programs, "gen_params" and "gen_coords", are available for input parameter generation and coordinate generation, respectively. In addition, an auxiliary program for sequence generation is provided, which is further discussed in the Supplementary Information. In the following two sections we explain the algorithm and ideas behind parameter file generation and system coordinate generation.

**Parameter file generation**. The problem of generating parameter files is treated as a graph transformation within polyply. Graph transformations are commonly used in other tools for generation input parameters as well[19,29]. The graph transformation in polyply takes a residue graph and maps it into a higher resolution graph, which is agnostic to the target resolution. A graph consists of nodes and edges. Edges describe which nodes are connected and nodes can have attributes that store specific information. In

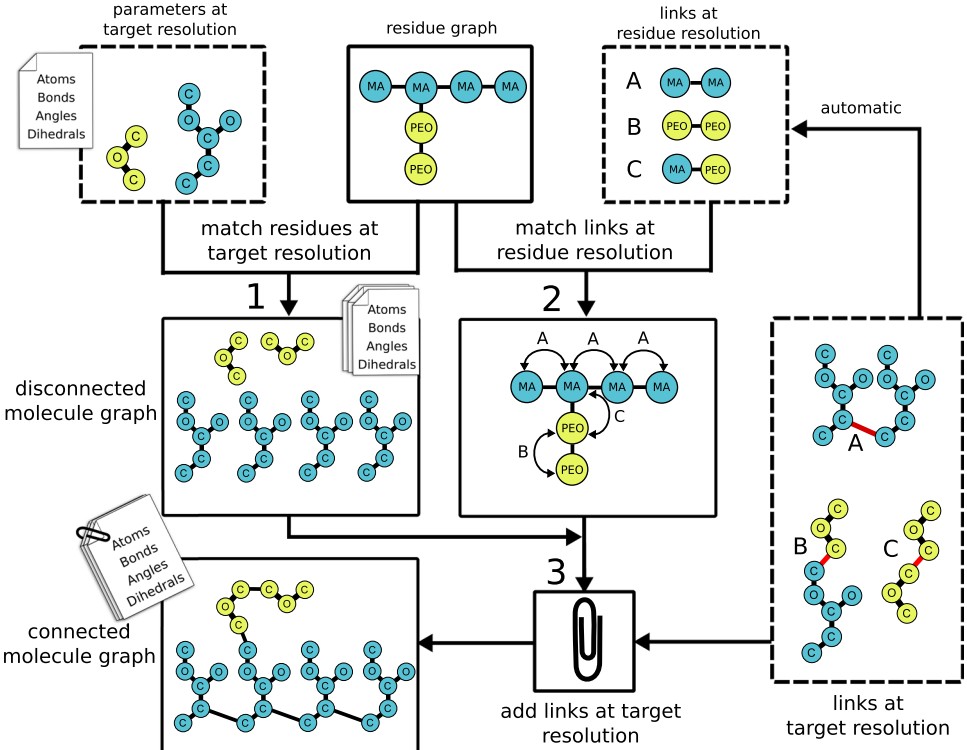

**Fig. 1 Schematic illustration of the workflow for parameter file generation.** Polyethylene oxide (PEO) grafted methyl acrylate (MA) is used as an illustrative example. The user input is the residue graph, while the building blocks are taken from the library (presented in dashed boxes). However, users can modify the library as they need. In step one, the parameters for the blocks are applied based on the residue graph to form a disconnected graph at the target resolution. In step two the links are matched at residue graph level to the input residue graph. Subsequently, links are applied to the disconnected graph at target resolution in the third step producing the complete parameter file.

this context a graph representation of a molecule translates the connectivity of bonds to the edges of the graph. Molecular characteristics of the atoms (e.g., their name or residue name) are stored as node attributes. Before we detail the algorithm, it is handy to define a few more terms: We define a block to be a graph, which corresponds to all interactions and atoms of a single residue; Complementarily, a link describes the interactions (e.g., bonds or angles) introduced when two residues are connected. To this end, polyply internally uses the Networkx[43] and vermouth[44] python libraries for handling graph-related computations.

The general input to polyply for parameter file generation is a residue graph of the target molecule. In addition, the blocks and links corresponding to the residues in the target molecule are needed (cf. Fig. 1, dashed boxes). Currently, polyply is shipped with libraries containing these parameters for some force fields and polymers, a database that will be expanded over time. From the definition of the residue graph, blocks, and links, polyply generates a parameter file in three steps (cf. Fig. 1):

Step 1: Generate a disconnected graph of residues

After reading the input files, polyply iterate over all residues of the input residue graph. For each residue, the matching block is added to an empty graph thereby generating a disconnected graph of residues at the target resolution. This graph already contains all atoms of the target molecule and interactions within the residues. Thus, the problem to assign the proper interactions linking the two or more residues remains.

Step 2: Find all links at the residue level

To generate all interactions spanning more than one residue, links are applied between two or more residues. To solve this in a general manner we treat it as a subgraph isomorphism problem at residue graph level: we find all the ways a link can fit onto the residue graph subject to constraints such as matching node

attributes. Performing this on the residue graph drastically reduces the problem size compared to solving the subgraph isomorphism problem at the target resolution. This establishes at residue level which links apply between which residues.

Step 3: Matching generic links to specific residues

Taking the matches between links and residues, the program establishes a correspondence between the atoms of the link and the atoms in the disconnected graph at the target resolution. To do so the atom names and relative residue indices given in the link are simply matched to the atoms of the residues in the disconnected graph generated in step 1. However, this matching step is not limited to the atom name and residue index. It can also be extended to take other atom characteristics into account. This allows accounting for information that is not encoded in the connectivity of the residue graph, such as chirality or anomers of the same residue. When a link is added, also the edges of the link are added to the disconnected residue graph. In this way, the disconnected graph gradually becomes a connected graph at the target resolution level. This completes the graph transformation and the molecule including all interactions only needs to be written to a file.

**System building**. Starting coordinates for systems are built using a generic multiscale approach, in which first a super CG resolution representation of the system is generated, followed by a back transformation to the target level. This multiscale approach is similar to the procedure underlying the Charmm-GUI polymer builder[30], but our approach is generic meaning parameters of the super CG model are derived on the fly based on the target force-field, employs a self-excluding random-walk in contrast to a full-scale dynamics simulation, and uses an automated back

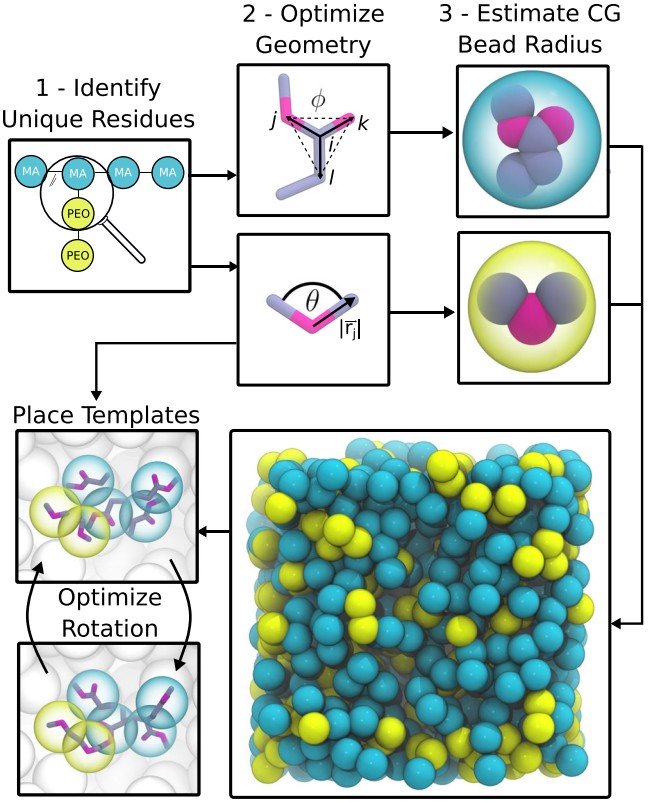

**Fig. 2 Principle of the multiscale algorithm for building systems.** The algorithm works in five steps: first residues are identified, followed by generating coordinates for the residues through two-step graph embedding. Based on the residue volumes the generic CG model is generated, which is then used in a self-excluding random-walk. Finally using the template residue coordinates the CG model is back mapped to the target coordinates.

transformation, which does not rely on a library of coordinate fragments. The system-building proceeds in five steps (cf. Fig. 2):

Step1: Mapping all molecules to one bead per residue

In the first step, the topology file is analyzed and all molecule types in the system are detected. For each molecule, all unique residues are identified and converted to blocks. A generic one bead per residue super CG model is created and stored in the form of a graph. The underlying connectivity of the residue graph is extracted from the bonded graph of the molecules.

Step 2: Generate coordinates for residues

Each block is a graph of a single residue and graph embedding is used to generate coordinates for this residue. Due to the specific requirements of molecular geometry, we utilize a two-step graph embedding. First, initial coordinates are generated using the Kamada-Kawai[45] embedding as implemented in the NetworkX library[43]. Subsequently, we perform a geometry optimization based on the bonded interactions within the residue using the scipy[46] implementation of the Limited-memory Broyden–Fletcher–Goldfarb–Shanno minimizer[47] (see Supplementary Fig. 1A for a detailed workflow). As presented in Supplementary Note 3 benchmarking on the residues in the library shows that this procedure yields reliable coordinates for residues.

Step 3: Derive parameters for the generic CG model

In the self-excluding random walk, a one bead per residue approximate CG model is used. It is based upon a Lennard–Jones (LJ) potential as the interaction function. The epsilon parameter

of the model (LJ well depth) is always set to 1 kJ/mol. Since we do not perform dynamics the attractive part of the potential is less important. The sigma parameter, however, determines the overall packing density and is computed from the residue template coordinates, reflecting the volume of the residue. It is derived from the radius of gyration as detailed in the Supplementary Information. In general, the radius of gyration is often used in polymer physics to estimate the spherical volume occupied by a single chain. Here, we port this concept to the molecular geometry of a single residue. However, in addition to the geometry of the residue, we also account for the fact that the single atoms have a certain volume. Although it is an approximate radius and probably not useful in an actual simulation, we find it to be good enough in the context of the random walk.

Step 4: Constrained random walk

To generate coordinates for the one bead per residue molecules in our target system, we perform a self-excluding random walk. An attempt to place a bead (step) in the random walk is rejected, if a maximum force on the placed bead is exceeded. The self-excluding random walk is by default performed along a breadth-first traversal of the molecular graph. This means nodes (i.e., residues) that are close to each other are placed first and then the algorithm proceeds further along the chain. Molecules are placed separately after each other, where the starting point is randomly chosen from a grid.

This grid can either be user-specified or is considered to be rectangular across the box. When the random-walk algorithm exceeds a certain number of steps, it goes backward in the breadth-first path by default by ten residues and tries to replace these ten residues. In addition to the force-acceptance criterion, the random-walk can also be supplemented with three additional conditions which help steer overall conformations: (1) geometrical constraints can be set to define regions of space that are excluded from sampling; (2) dimension constraints can be specified to limit the random walk dimension along specific vectors to for example create brushes on surfaces; (3) distance restraints can be set to define target distances between specific nodes within a molecule. To meet the distance restraints, polyply implements a graph-based algorithm that puts upper and lower bounds on each step taken. Details and benchmarking are provided in Supplementary Fig. 2 and Supplementary Note 4, subsection 2. All additional conditions have to be specified in a build file. The file syntax for the build file is available online (https://github.com/marrink-lab/polyply_1.0/wiki/Syntax:-build-file). Furthermore, to generate starting structures for polymers with high persistence lengths, polyply implements a feature that lets the user set a persistence length. Subsequently, the program samples from the end-to-end distance distribution of the worm-like chain model and use the distance restraint algorithm to set the end-to-end distance (see Supplementary Note 3 for more details). All interactions are computed considering rectangular periodic boundary conditions using the scipy c-implementation of the KD-tree[46]. Using a KD-tree makes it possible to compute a large number of distances within a cut-off efficiently within python.

Step 5: Backmapping

Low-resolution coordinates are transformed to the higher resolution target coordinates by a residue template-based back-mapping procedure (Supplementary Fig. 3A) similar to those used for biomolecules[48]. First, the center of geometry of the residue template is moved to the CG position. Subsequently, we optimize the rotation of the template around the center of geometry such that atoms that have a bond to other residues are placed close to those residues. To do this we first perform a connection analysis finding which atoms connect to atoms of the

neighboring residues. If the target resolution coordinates are available, they are used over CG coordinates. Overall, this aligns the residue template with the chain backbone in a generic fashion. This procedure is also applicable to branched residues and optimizing the rotation is efficiently done in python. Similar to the idea in the backward program[49] the template residue coordinates are scaled by a fudge factor of 0.45 from the center of geometry. In the final step, the user has to run a regular geometry optimization. This causes the coordinates to relax to a final state that can be used as input to run an equilibration simulation. As discussed in Supplementary Note 3, subsection 3 this procedure is efficient in avoiding high energy states as well as artifacts such as interlocking rings.

To implement chirality, we currently use a specific improper dihedral that forces chirality during the energy minimization step and/or during the template generation.

In the following sections, we will demonstrate the capability of the program with some examples for realistic systems. In general, we will briefly present the steps required to generate both input parameter files and coordinates. For each system, we ran a small simulation to confirm the parameters are correct and the system is stable.

**Polymer melts at atomistic and CG level**. Simulation of amorphous polymers or melts is the backbone of much research in material science and polymer science in general. Depending on the length and time scales that are needed atomistic resolution or CG models are utilized. Here, we demonstrate that polyply is capable of generating realistic melt conformations independent of the target force-field, and we analyze its performance. In particular, we have generated melt systems of varying sizes for poly methyl acrylate (PMA), polystyrene (PS), poly methyl methacrylate (PMMA), poly vinyl alcohol (PVA), polyethylene (PE), and PEO. For each system, we generated the required parameter files and initial coordinates for 100 chains of length 50, 100, 250, and 500. Each of these systems was generated at an atomistic level using GROMOS and CG level using the Martini3 force-field. Figure 3a shows the atomistic structures of the six polymer species surveyed and overlaid as circles those atoms treated as one particle in the Martini CG models. After system generation with polyply, we ran an energy minimization and computed the average end-to-end distance of the polymers from the minimized configuration. To improve statistics, ten replicas of each system were generated leading to a total of 480 systems. The initial target densities are reported in Supplementary Table 3.

In a melt, polymer conformations can be described by ideal chain statistics. Some properties like the end-to-end distance can be computed from theory utilizing experimental input quantities such as the characteristic ratio. To show that the initial polyply conformations are realistic for melts, we computed the expected end-to-end distance using two models—the hindered rotation model (HRM) and the worm-like-chain model (WCM)—and compare those distances to the end-to-end distance generated by polyply. Figure 3b shows the comparison. Overall, the polyply structures clearly follow the trends of both models, but also the quantitative agreement is good with mean absolute errors of about 5.8 Å compared to the HRM and 9.5 Å compared to the WCM. For the WCM we also compared the distributions obtained from polyply and the model as shown in Supplementary Fig. 4. We find a good qualitative agreement.

It should be noted that of course the density and end-to-end distance of the force field can be different from experiment or theory. The end-to-end distance can be further fine-tuned by optimizing the step length of the random walk. However, even if not optimized, relaxation to target density and end-to-end distance is typically observed within 50 ns or less, as shown in Supplementary Fig. 5 for a subset of systems. Figure 3c shows a single chain of PS 100 in a melt after 5 ns of simulation. Other chains are removed within a 1 nm radius around that chain to illustrate the coil-like conformation of the chain in the melt. Figure 3d shows the typical time it takes to generate coordinates for different numbers of total residues using polyply. Generating CG residues is slightly faster than generating atomistic residues. Overall melts of up to 50,000 residues are typically generated in less than 5 min. Larger systems in the order of 1 million residues are generated within less than 70 min.

**Single-stranded DNA and circular single-stranded DNA**. DNA is an important bio-macromolecule that expresses and regulates genetic information in cells. Whereas most of the genetic information is encoded in double-stranded DNA (dsDNA), single-stranded DNA (ssDNA) is frequently important in replication and repair processes. Beyond this, there are also a number of DNA viruses that encode their genetic information in ssDNA, and ssDNA patches are for instance found in telomers[50]. Generating realistic structures for ssDNA provides another level of challenge to the multiscale random-walk protocol. On the one hand, DNA nucleobases are large residues with about 30 atoms at the all-atom level. On the other hand, the persistence length of ssDNA is about 3-10 times higher than that of most flexible polymers[50]. Finally, ssDNA is highly charged and known to coordinate ions around the chain increasing the persistence length[50]. To verify that polyply is able to generate configurations for these macromolecules, we implemented the Parmbsc1 force-field[38] into polyply and built a distribution of poly-T ssDNA with different lengths (8, 16, 50, 65, 100 bases). For each length, 100 replicas were generated. The persistence length of DNA is known to change as a function of the salt concentration[50]. For each chain length, we, therefore, set two experimentally determined persistence lengths (3.2 nm, 1.4 nm) corresponding to low (12.4 mM/L) and high (1 M/L) salt concentrations. This resulted in 1000 DNA structures generated by polyply. Figure 4a shows the radius of gyration as a function of chain length for the two sets of persistence lengths compared to SAXS values. Note that the random-walk protocol is not biased against the radius of gyration. Agreement between the generated structures and the experimentally measured values is good. A further characteristic of polymer conformations frequently applied in polymer physics is the scaling of the radius of gyration with the number of monomers following the equation ($R_G = A \times N^v$). The scaling measured with polyply at low salt concentration ($v = 0.71 \pm 0.01$) matches the scaling found in experiment ($v = 0.72 \pm 0.01$) whereas the scaling at high salt concentrations ($v = 0.55 \pm 0.01$) is somewhat lower but still close to experiment ($v = 0.57 \pm 0.02$). Figure 4b shows two DNA chains with (100 bases) corresponding to the average radius of gyration seen in Fig. 4a. The principle components of the chains were aligned showing that both are extended but the chain with higher persistence length clearly is more elongated. To further investigate how good polyply DNA structures are a starting point for AA simulations, we ran a simulation of poly-T at the all-atom level and low salt concentration. As shown in the Supplementary Note 5, subsection 2 the end-to-end distance and radius of gyration distribution obtained in an unbiased simulation overlap with those generated by polyply. Finally, we generated coordinates for the full genome (1767 bases) of the porcine virus within the virus capsid at the all-atom Amber level. First the known genome sequence[51] was converted to a circular graph and then used as input to polyply

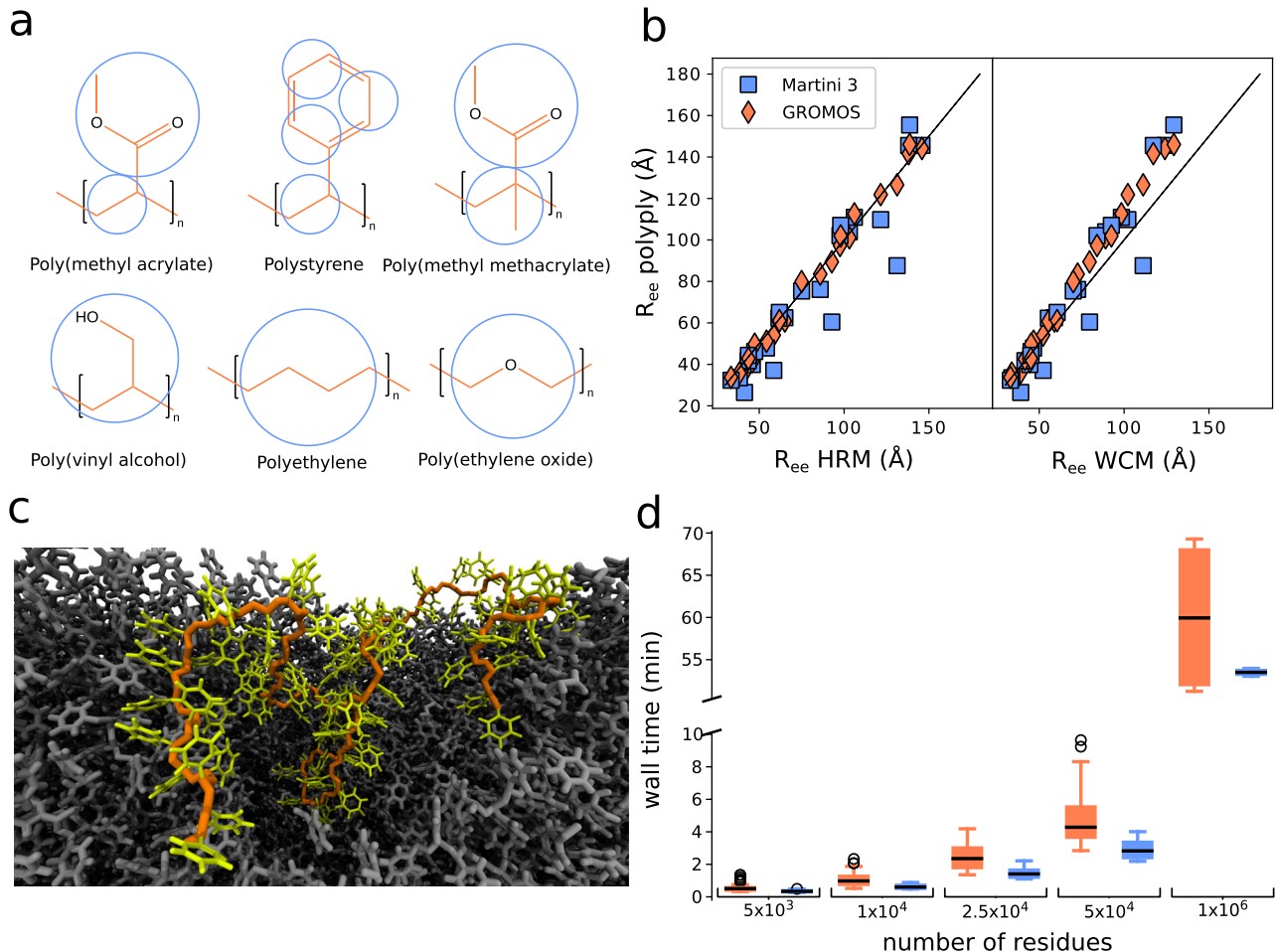

**Fig. 3 Characteristics of melt systems generated by polyply and performance. a** Atomistic structure (orange) and mappings to Martini level (blue) for all six polymer species surveyed. **b** End-to-end distance of the melt structures generated with polyply compared to those obtained by theory using the HRM (left) and WCM (right). Blue squares indicate Martini structures, and orange diamonds GROMOS. For each of the polymers 10 systems with four different chain lengths (50, 100, 250, 500) were built. **c** Single chain of atomistic Polystyrene with 100 residues shown in the melt, other chains are shown in gray and residues within 1 nm are omitted for clarity The backbone of the chain is highlighted in orange and the side chains in yellow. **d** Typical time for generating coordinates with polyply for different total numbers of residues. The performance is shown per force-field (blue: Martini3, orange: GROMOS), averaged over all systems in panel **b** leading to a total sample size of $n = 60$ for all systems with less than 1 million residues. In addition, systems with 1 million residues have also been set up as a further benchmark, however, considering only five replicas (i.e. $n = 5$). The boxplots are shown as the median within the bounding box corresponding to Interquartile-range (IQR) (Q3–Q1) and Tukey-style whiskers extending to 1.5 IQR. Open circles are outliers and in the case of atomistic melt systems, the outliers are all from melt systems of Polyethylene. Source data are provided as a source data file.

gen_params to generate the input parameters. Subsequently coordinates were generated for the entire system using polyply gen_coords. A detailed protocol can be found in Supplementary Note 5, subsection 5 and Supplementary Fig. 7. Important to note is that the coordinates of the capsid[52] have to be supplied as input to polyply together with the constraint that the ends of the DNA chain must be in contact (i.e., circular) and that one sodium must be placed near each DNA residue to ensure the neutrality of the DNA within the capsid. Afterwards the system was relaxed and we ran 60 ns of equilibration of the entire system with no restraints on the DNA to ascertain that the system is stable.

**Polymeric lithium-ion battery.** Ion conducting polymers for the application in lithium-ion batteries has been a very active field of research for many years[11,35,53]. Block-copolymer systems, consisting of one conducting polymer and one polymer improving the mechanical stability, are a very promising route to new and enhanced batteries. Simulation of ion conduction in these systems, however, are less common as creating the initial coordinates

poses several technical challenges: (1) The system has to be obtained in the phase-separated state. Modeling salts in common bead-spring models or mean-field theories is difficult when the effect on phase separation is unknown. (2) The salt has to be distributed within the PEO layer but without generating overlaps. Especially, for the commonly used large anions this becomes problematic. (3) In principle, the material is best represented by a multilamellar system, in which for example cross conduction can also be observed even though for PS-b-PEO this is known to be less problematic. In this example, we will show how to create a multi-lamellar system of PS-b-PEO doped with lithium bistriflimide (LiTFSI) using realistic polymer length as well as representing the ions explicitly. Our target system comes from the experimental work of the group of Balsara, who has exhaustively studied PS-b-PEO doped with LiTFSI[54].

The block-copolymer in this example consists of PS with a molecular weight of 6.4 kg/mol (~63 monomers) and PEO with a molecular weight of 7.3 kg/mol PEO (~163 monomers). LiTFSI is mixed in using a ratio of Li to PEO monomers of 0.085.

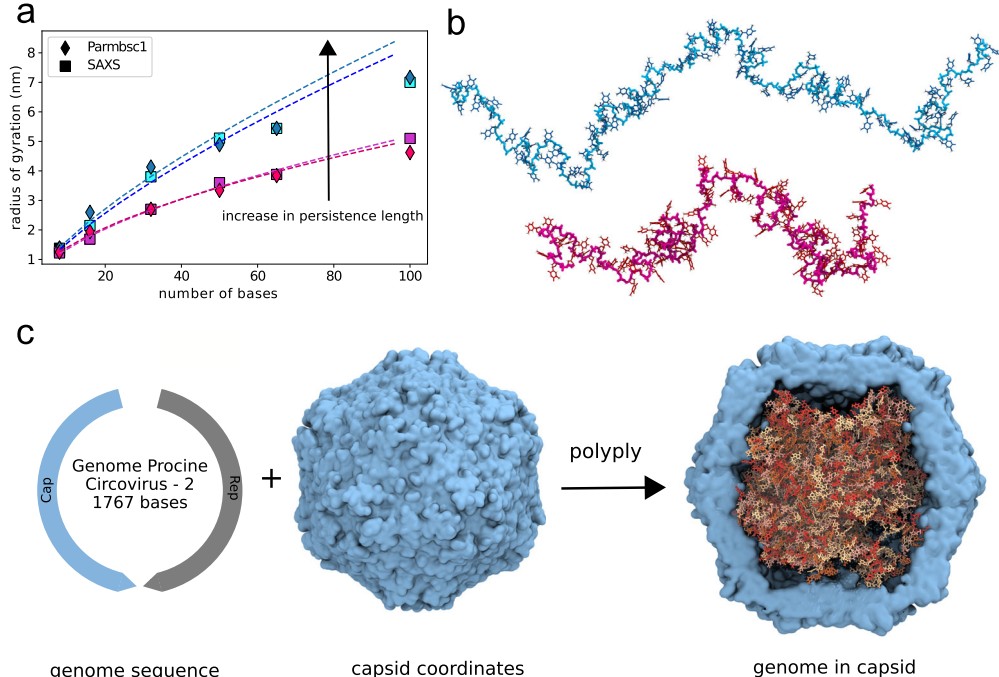

**Fig. 4 Single-stranded DNA test case. a** Average radius of gyration from an ensemble of polyT generated with polyply using two different persistence lengths (1.4 nm; 3.2 nm) compared to the radius of gyration from SAXS. **b** polyT chains of 100 bases corresponding to the average radii of gyration in (**a**). The principle components have been aligned. **c** Schematic of generating circular ssDNA for the full genome of the porcine virus. Final coordinates are shown on the right with the DNA colored by nucleobases and the virus capsid in cyan. The image of the genome sequence was adopted from ViralZone SIB Swiss Institute of Bioinformatics. Source data are provided as a source data file.

From the experimental work it is known that this composition forms a lamellar phase with a domain spacing of 20 nm. The Martini input files for the polymers are simply generated using polyply gen_params and our Martini library of polymers. To generate the starting structure in the phase-separated state, we specify rectangular geometric restrictions on where each part of the block-copolymer is allowed in the box. To that end, we define six alternating domains of 12 and 6 nm sizes in which PEO and LiTFSI or PS are allowed, respectively. The domains are unequal because the volume fractions of PS and PEO are not the same. To provide the Martini consistent domain spacing, we generated a system with a single lamella and equilibrated the volume. The thin film is generated in two steps, aiming at a uniform salt distribution in the PEO part. In a first step, the salt is dispersed throughout the box placing it inside the domains where PEO is allowed. Figure 5a shows the salt as well as the domain boundaries as obtained after this step. Subsequently, we generate a grid of starting points on the boundary of these domains. Starting on these grid points the chains are grown into the domains by our random walk using the generic super CG model. This approach has previously been shown to be adequate for simple bead spring models as well[34,55]. Finally, the program backmaps the structure to Martini's target resolution. Once the starting structure is generated, which typically for this size of system takes 30 min, energy minimization is performed. Subsequently a short equilibration of 5 ns keeping the z-dimension fixed and only applying pressure coupling in *xy* is performed to allow for chain packing to increase orthogonal to the stack direction. Subsequently we ran a 50 ns equilibration under constant area only coupling the z-direction to equilibrate the salt distribution. Figure 5b shows the morphology of the system after this short equilibration phase. The thin film has a size of about 60 nm × 60 nm × 10 nm and comprises roughly 600,000 particles. As shown by the zoomed view as well as the density

profiles (Supplementary Figure 9), the space is completely filled with polymer and salt.

**Lipid vesicle with liquid–liquid phase separated interior**. Liquid-liquid phase separation (LLPS) is an important driving force in both biotechnological applications and biological systems. Systems capable of undergoing LLPS are therefore of high interest to many researchers and are not only studied experimentally but also at various levels of theory[10,56]. Concerning cellular processes, LLPS is speculated to have promoted the early stages of life by allowing to form simple compartmentalization, which eventually lead to the evolution of membrane-less organelles inside modern day cells[57,58]. As such, studying LLPS in the context of cellular environments is of considerable interest. While the supporting programs for bio-molecular simulations allow the generation of cell membrane structures of entire mitochondria or virus envelopes, filling those with anything else than water and ions is usually challenging, especially when polymer phases are to be simulated. In this example, we set up a system consisting of a multicomponent lipid vesicle, composed of dioleoyl-phosphatidylcholine (DOPC), dipalmitoyl-PC (DPPC), and cholesterol, and containing PEGylated 1-palmitoyl-2-oleoyl-phosphatidylethanolamine (POPE) lipids, filled with a phase-separated aqueous solution consisting of PEO and dextran in the interior. This system has been experimentally shown to induce vesicle fission and therefore gives insight into the generation of early life[59]. Similar systems are also considered important to the development of synthetic cells[60].

This example not only demonstrates that polyply interfaces well with the area of biomolecular simulations, but also demonstrates several technical challenges: (1) PEGylated lipids require the addition of PEG to the lipids of the vesicle and need to be placed without penetrating the bilayer; (2) dextran is a branched sugar polymer, which typically has a statistical distribution of molecular weights and

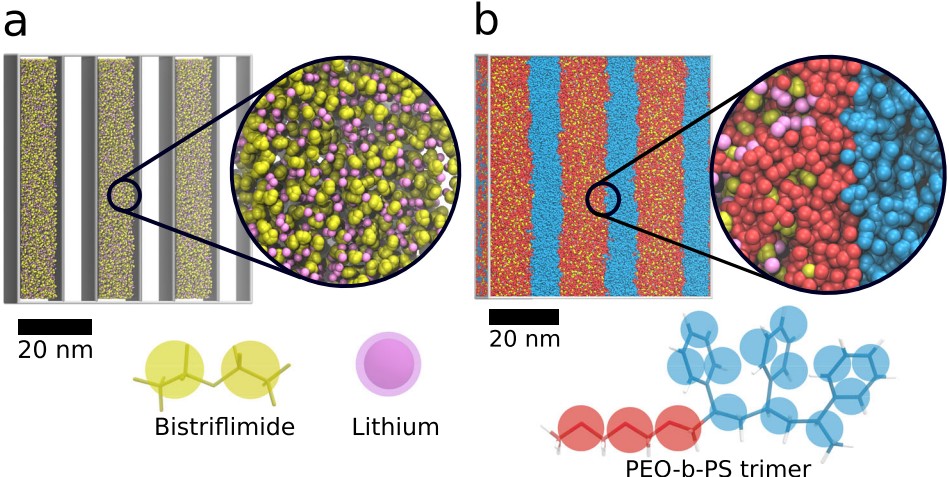

**Fig. 5 Phase separated block-copolymer PS-b-PEO doped with lithium bistriflimide (Li-TFSI) generated with polyply. a** Li-TFSI salt placed within the domains where PEO is going to be located. The zoom provides a more detailed view of the salt also showing that there is empty space in the initial dispersion. **b** Structure obtained after growing in the block-copolymer around the salt shown in panel A and running 50 ns of equilibration. The generated morphology contains six lamellar regions. The zoom shows an in-depth view onto the interface between the two polymers, showing that empty space has been filled. Below both panels, the mapping of the all-atom residues to the CG level is shown.

branches; (3) the coacervate needs to be generated in the phase-separated state. To overcome this challenge, first, parameter files for PEGylated lipids and PEO are generated using the Martini library and the gen_params tool. The molecular weights of PEO in the vesicle and in the PEGylated lipids are 2000 g/mol (~45 monomers) and 8000 g/mol (~180 monomers), respectively. Generating parameter files for dextran is more complicated, however. Dextran is a polysaccharide composed of α-1,6 linked glucose residues with α-1,3 connected branches. In addition, it is in general polydisperse and the branching depends on the molecular weight. For the target molecular weight of 10,000 g/mol, dextran has on average 5% branches from the main chain with a length of up to three residues[61,62]. To model the diversity in dextran's molecular structure we generated 500 residue graphs, with random number of branches and lengths as outlined in the Supplementary Notes 5, subsection 4. The whole workflow used to generate the system is described in Supplementary Fig. 10. Using this distribution of structures (Supplementary Fig. 11), polyply gen_params was used to create parameter files for all those structures, which took less than 15 min.

To generate starting coordinates for this system, we first obtained a vesicle using TS2CG[24]. As there are more specialized programs to generate lipid bilayers in various shapes and forms it was not our intention to also generate those using polyply. The lipid coordinates generated by TS2CG were given as starting structure to polyply. In addition, a geometric constraint was used to specify that PEO and dextran can only occupy half of the vesicle, approximating it as a sphere, with a region of 2 nm overlap to allow some interphase mixing. With this input, the system is generated by our generic super CG random walk followed by a backmapping step. Polyply also automatically identifies that PEGylated lipids have to be extended, as the random walk algorithm will steer the conformations away from the membrane. Generating the entire system took about 30 min. Once the initial coordinates are obtained, an energy minimization was run, followed by a short equilibration and 50 ns of production run. The system was stable, and its final configuration is shown in Fig. 6. We clearly see that the system remains phase-separated at least over the timescale simulated and that the PEGylated lipids are uniformly distributed both on the inside and outside of the bilayer.

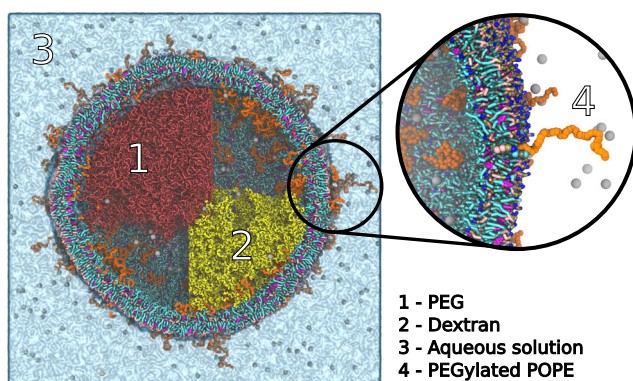

**Fig. 6 Liquid-liquid phase separation inside a vesicle.** The vesicle is composed of DOPC, DPPC, cholesterol, and PEGylated POPE (PEG part in orange), containing a coacervate of PEO (red) and dextran (yellow). Inside and outside the vesicle are water and sodium to counterbalance the negatively charged PEGylated POPE. The diameter of the vesicle is 40 nm and the left half is filled with PEO and the right with dextran. Some polymers in the upper right and lower left are omitted to show the ions and PEG tails of the PEGylated lipids on the inside. The zoom shows more detail of the membrane showing a PEGylated lipid extending from the bilayer surface.

## Discussion

The increase in computing power propels MD simulations of systems into new areas, which in their complexity or size were unmanageable only a few years ago. However, equilibration of such systems far from the desired equilibrium state takes time and is costly in terms of computing power. A second complicating factor relates to generating input files for such simulations, as setting up the input parameters not only takes human time but is also intrinsically error-prone. The latter aspect becomes especially problematic when HT approaches are being designed. Programs which facilitate HT MD simulations featuring multi-component complex systems are therefore highly desirable. While those exist for biomolecular simulations, they are largely missing in the field of material science. To resolve this situation, we have

defined five major challenges a program needs to be able to solve: (1) it needs to be resolution and force-field independent; (2) able to set up complex and large systems; (3) support arbitrary complex molecular topologies (i.e., branching); (4) be able to generate complex morphologies; and (5) and finally be reasonably fast to permit HT research.

In this paper, we have presented polyply, a software suite that is aimed at facilitating simulations involving polymers at any target resolution and force-field desired. Parameter file generation and coordinate generation are split into two independent programs—polyply gen_params and polyply gen_coords. The gen_params program implements the graph transformation used within polyply to generate parameter files. The algorithm takes a residue graph and maps it into a higher resolution graph, which is agnostic to the target resolution thereby meeting the first of the challenges outlined above. It further matches fragments, which describe the bonded interactions between residues, by finding all subgraph isomorphisms between these fragments and the residue graph provided. Therefore, it will also assign parameters for complex branched or cyclic polymers. This has been shown to work based on our test cases. Especially the last test case exemplified this aspect, where polyply was used to generate a statistical distribution of the branched dextran polymer that involves two different linkage types between otherwise equivalent residues. As outlined in the challenges, polyply is also applicable to bio-macromolecules, which we have shown previously by using polyply as part of a protocol for setting up PEGylated proteins[63] and glycans on the SARS-CoV2 spike protein[64]. We also demonstrated that polyply is capable of generating input parameters for single-stranded (circular) DNA, further strengthening our claim that the program is applicable to a wide range of biopolymers.

In addition to supporting large and complex polymers, generating input parameter files is also fast, making it suitable for HT applications. For example, generating a parameter file for an atomistic polystyrene chain of 1000 residues, which involves more than 100,000 bonded interactions, takes less than 10 s. Overall, we showed that the polymer parameter generation meets all requirements outlined in the challenges. Currently, the main limitation of the parameter generation is that it is limited to GROMACS input files. However, extension to other MD software is possible and would only require additional input file-parsers as the core of the code makes no calls to the GROMACS software itself. Meanwhile, interested users can already use existing input file converters[65] to connect polyply to other MD engines. Supplementary Table 2 provides an overview of conversion programs.

As further outlined in the challenges, structure generation is a key step for any program. It needs to be fast and generate complex structures that are good enough to start the simulation without extensive relaxation. To this end, polyply gen_coords implements a multiscale approach to generation of condensed phase systems at near to target density. The multiscale approach is based upon using a generic one bead per residue CG model, performing a self-excluding random walk, and backmapping it to target resolution. The interaction potential used for the self-excluding random walk, which essentially approximates the volume of a residue, is directly computed from the geometry of the residue as well as the force-field parameters of the target polymer. Thus, our multiscale approach can be used independent of the force-field and target resolution. To further validate the robustness and quality of this approach, we generated 480 melt systems at two levels of resolution for four polymer species. For all systems an energy minimization could be run without failure. Not only were the systems stable, but the average end-to-end distance of the initial frames also averaged over ten replicas

compared very well to theoretical calculations for melts. This suggests that the distribution of conformations and entanglement produced by polyply are realistic, providing a good starting point close to the equilibrium configuration. This is further confirmed by running test simulations for some replicas, which show little relaxation of the end-to-end distance to the force-field specific value as well as to the force-field specific target density. Whereas classical polymer melts consist of relatively flexible polymers, especially biomacromolecules such as DNA can have long persistence lengths. By generating ssDNA configurations setting different persistence lengths and comparing to experimentally available SAXS data, it was demonstrated that polyply can also handle stiff polymers in a satisfactory fashion. Furthermore, the multiscale coordinate generation uses a random walk breadth-first traversal of the molecular graph. This means residues which are connected by bonded interactions are placed close in space, making it possible to generate even complex branched structures.

Using a one bead per residue CG model not only makes the approach force-field and resolution-independent, but it also greatly increases the performance. We assessed the performance for 480 melt systems up to 50,000 residues. For polystyrene at Martini level 50,000 residues equates to 200,000 coordinates and at GROMOS level to 600,000 coordinates. These systems are typically generated in less than 5 min independent of the chain length and target force-field. This performance compares favorably to the recently published PyPolyBuilder, which takes about 8 min for a single chain of 1248 coordinates[29]. To further benchmark the performance of the structure generation, we also set up systems with 1 million residues, corresponding to 13 million coordinates for atomistic polystyrene. These are generated on average within less than an hour on a single core.

As the generic multiscale approach is very efficient in packing even long polymer chains, it was possible to augment it with further acceptance criteria to steer the random walk chain placement. For example, it is possible to force the direction of the random walk, which makes it very suitable for generation brushes. An example is outlined as part of our online tutorials. In addition, simple geometrical constraints can be utilized to build phase-separated systems. By generating a liquid–liquid phase separated system consisting of highly branched dextran and PEO inside a vesicle as well as a micro phase-separated block copolymer PS-b-PEO system, we have shown that these tools can generate a variety of inhomogeneous systems. The former example also shows that it can easily be combined with already existing systems, especially those for which more specialized builders exits, such as lipid membranes. Finally, we showed an implementation of a distance restraint algorithm that lets users define target distances between nodes. This enables polyply for example to create large macrocycles such as circular DNA, as demonstrated by generating the complete circular genome of the porcine virus. Furthermore, this algorithm in principle enables users to take experimental information into account. For example, distances from NMR experiments could be converted to distance restraints.

Overall, these examples demonstrate that polyply is capable of setting up large and complex systems, with an excellent performance in a realistic but force-field and resolution-independent manner. However, the coordinate generation also has some limitations. Custom non-bonded or bonded interactions that deviate from standard bonded interactions or the default pairwise LJ interactions are currently not taken into account. It is not possible to generate coordinates for polymers that have a well-defined geometry such as double-stranded DNA, neither at the all-atom or CG level. As DNA geometry is well defined, we are working on an extension that would allow us to generate dsDNA taking those

constraints into account. A further limitation is that currently the structure generation only supports rectangular PBC conditions. Although they are sufficient for a wide variety of systems, more complex PBC conditions could be implemented by extending the scipy cKD tree, which is the workhorse for interaction computations. Finally, while simple geometric restrictions for the random-walk work fine for many applications, extending them to arbitrarily shaped boundary surfaces would enable to directly read in experimental density maps and then grow polymers on top of those. This would require some form of triangulated surfaces to be used as boundary surfaces, as is done in TS2CG[24].

In conclusion, we have demonstrated that the software suite polypy, presented here, is able to generate input files and starting coordinates for complex and challenging systems, connecting the bio-molecular world to material science.

## Methods

**MDs settings and analysis**. All MD simulations were done with GROMACS[66] (versions 2020 and 2019), using the Verlet cut-off scheme. For the CG simulations, a cut-off of 1.1 nm was used for the LJ interactions, whereas the Coulomb interactions were computed within a cut-off of 1.1 nm, and PME for longer-range contribution to the electrostatics whenever the system at hand contained charges. The energy minimizations were done with the standard steepest descent algorithm as implemented in GROMACS. The CG MD simulations were performed using the default leap-frog integrator with a time-step of 20 fs in the isobaric-isochoric ensemble. The temperature was kept constant using the v-rescale algorithm[67] and the pressure was coupled using the Berendsen barostat[68], typically used for equilibration of MD simulations.

The atomistic MD simulations with the GROMOS 2016H66 force field were conducted with a cut-off of 1.4 nm for the LJ interactions and electrostatic interactions. Long-range electrostatics were further treated with the reaction field method, where the dielectric constant was set to 2 for all polymer systems, which is reasonable considering the typical low dielectric constant of vinyl polymers. Atomistic simulations with the Parmbsc1[38] and Amber[39] force-field used a cut-off of 1.0 nm for both LJ and electrostatic interactions. Long range electrostatics were treated with PME. Energy minimizations were conducted as well with the steepest descent algorithm and MD simulations were run with the default leap-frog integrator using a time-step of 1 fs for the integration. The time-step of 1 fs is necessary as the relaxation runs to target density are done using unconstrained bonds. For production runs the time-step can be increased to 2 fs with constrained bonds, in principle. As explained for the CG simulations also the GROMOS simulations were run in the isobaric-isochoric ensemble using v-rescale[67] temperature coupling and Berendsen pressure coupling[68]. Analysis of the end-to-end distances and radii of gyration was done using MDAnalysis[69] and gmx polystat[66]. Fitting of the radius of gyration data was done using the symfit (v.0.4.6) under consideration of the standard error. Snapshots of simulations were prepared with VMD[70] (v. 1.4.9a) and figure complied with Inkscape (v.0.52.0) and matplotlib[71] (v.3.3.4).

**Systems**. Polyply was run for all test-systems within python3 (v3.6.9 on local machines, and v3.8.2 on the Dutch National Supercomputer, Cartesius). For acceleration, the numba package[72] was installed in all environments. All test systems were either run on a local desktop machine running Linux OS or on the national HPC cluster. The performance for parameter file generation and input coordinate generation were recorded on the HPC cluster for the CG systems, running 24 processes in parallel on a single node. On the other hand, atomistic benchmark times were recorded on a desktop machine running ten processes in parallel.

*Models*. The all-atom polymer models were implemented following the rules for creating polymers within the GROMOS 2016H66 force-field and charges adopted from similar functional groups as is custom within GROMOS[40]. The current library does not only support homopolymers but all combinations of monomers. Parmbsc1[38] DNA parameters were adopted from the GROMACS implementation. The protein capsid Amber[39] parameters were obtained from gmx pdb2gmx[66]. Martini CG models were parametrized newly following the Martini 3 guidelines for making molecules or were adopted from existing Martini 2 models. Each new model was based on the GROMOS model, for which a system was generated with poyply and run with the settings stated above. We note that these models are subject to future improvement and that currently only homopolymers are supported with the exception of PS and PEO block-copolymers. A detailed validation of these models will be provided in a separate publication. The current (preliminary) versions are available from the polyply library (see code availability).

**Reporting summary**. Further information on research design is available in the Nature Research Reporting Summary linked to this article.

## Data availability
The polymer model parameters used in the examples are part of the polyply library (https://github.com/marrink-lab/polyply_1.0/tree/master/polyply/data). Input files and protocols for generating the example test cases can be found in the polyply regression test repository (https://github.com/marrink-lab/polyply_regression_tests/tree/main/examples). Tutorial input files are available from the same repository (https://github.com/marrink-lab/polyply_regression_tests/tree/main/tutorial_files). Source data are provided with this paper.

## Code availability
Polyply is distributed via the pypi package index (https://pypi.org/project/polyply/) and is developed publically on GitHub (https://github.com/marrink-lab/polyply_1.0) under the permissive Apache-2.0 license. Our wiki section on Github (https://github.com/marrink-lab/polyply_1.0/wiki/FAQs) provides an FAQ section for troubleshooting, as well as tutorials and additional information on file syntax.

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

## Acknowledgements

We would like to thank the Center for Information Technology of the University of Groningen for their support and for providing access to the Peregrine high performance computing cluster. S.J.M. acknowledges funding from the ERC via an Advanced grant "COMP-MICR-CROW-MEM", grant agreement ID 669723. Computational resources for this work were partly provided by the Dutch National Supercomputing Facilities through NOW and partly by the French National Supercomputing Center CINES via GENCI (grant A0060710138) awarded to L.M. The authors thank M. C. Ramos, Y.M.H. Gonçalves, and B.A.C. Horta, for checking the GROMOS input parameters and discussions on modeling polymers with the 2016H66 GROMOS force-field.; T.A. Mayer and A.J. Gormley for discussions on high throughput polymer synthesis and the future potential of simulations to enhance experimental high throughput protocols; Melanie König for her feedback and perspective on the figure design and the writing.

## Author contributions

F.G. conceived the project with suggestions from P.C.T.S., S.J.M., and L.M. F.G., P.C.K., and R.A. developed the code and algorithms and performed initial testing. F.G. and R.A. conducted and analyzed the MD simulations pertaining to the test cases presented in the paper and Supplementary Information with suggestions from P.C.T.S., F.G., P.C.T.S., and R.A. generated the polymer and molecule parameters. All authors contributed to writing the paper.

## Competing interests

The authors declare no competing interests.
