## [Peer Review File · Nature Communications]

REVIEWER COMMENTS

Reviewer #1 (Remarks to the Author):

In this work, the authors who have developed Martini CG models for long time are now developing general schemes to generate input files and realistic starting coordinates for complex bio and nano materials. Since MD simulations and molecular models have become matured now, modeling of complex starting structures in bio and nano materials are bottle necks in their simulations, as the authors say. Their Python suite, polyply, have several unique features, such as automatic generation of parameters for complex polymeric topologies, and a generic multi-scale random walk protocol for building structures. I think that this approach is interesting and useful for many applications in particular, polymer systems.

However, I have several concerns to be clarified before the publication of this work in this journal.

1) Force field independency of polyply

In the manuscript, the authors mention that polyply is applicable to nano and bio-polymers regardless of force field parameters. However, on page 12, they mentioned that parameter generation is limited to Gromacs input files only. For bio- and nano-material simulations, there are many other programs, for instance, AMBER, CHARMM, NAMD, LAMMPS, OPENMM, etc. If the authors indicate that this program is applicable for any force field parameters, they should prepare input file-parsers for other force-field as soon as possible. If this is not so easy, the authors should not emphasize the force-field independency in the manuscript.

2) Applicability for bio-polymers

For this reviewer, it is not so clear which biomacromolecules could be handled using this software. I would like to hear the current status in the program. I would like to hear which biomacromolecules are available using this software. I think that multi-scale random walk protocols for building structure is especially useful for polymer systems, while I wonder how it is applicable to bio-polymers, such as proteins, DNA, and RNA systems. Since MD simulations of cellular environments are one of the new trends, clarifying the applicability of complex biological systems using the software is meaningful for computational biophysicists.

3) Backmapping to build realistic starting structures.

In complex polymer systems, it is not straightforward to obtain good (low-energy) starting structures. Without special cares, starting structures could have many problems. Then, this requires long relaxation times or generates unstable trajectories in the MD simulations. I would like to hear

that in their backmapping, how they elaborate to avoid artificial unstable structures. For instance, entanglement of polymers or ring-penetrations are often pointed out as starting structures to be avoided. If the backmapping describing in Figure 2 includes any techniques, I would like to ask the authors extend such descriptions in main text or supporting information.

4) Spatial geometry of complex initial structures

In the manuscript, they showed three examples. Polymeric lithium ion battery and lipid vesicle with liquid-liquid phase separated interior are very complicated simulation systems. For my understanding, there is a big gap between the explanation in sections 2 and the examples in section 3. To reduce the gap, I suggest the authors to add schemes for modeling the two systems in supporting information, describing the modeling protocols step-by-step. Also, it is helpful to include the help of other software (like TS2CG) for the modeling. It should be clarified what is the input data and what is the output in each step.

5) Restraints from experimental information

In biomolecular modeling, inclusion of experimental information as restraint is essential. For instance, chromatin modeling is possible only when one incorporates the distance-restraints obtained from Hi-C experiments. Using polypy, can we introduce such restraint forces originally observed by experiments?

In summary, I believe that this tools even at the current stage could be regarded as a very useful tool in nano- or polymer sciences. If the authors would like to emphasize the applicability of this software in biological sciences, I would like to ask them to clarify the above points indicating the current functions or future perspectives.

Reviewer #2 (Remarks to the Author):

This is a really, really useful piece of work, which outlines a general framework and the corresponding workflows for setting up molecular dynamics simulations of polymeric materials. I anticipate the manuscript will be highly cited and widely used by the polymer community.

In terms of content, the actual methods or results are not new. The work focuses instead on describing a "tool" that will enable or facilitate research on polymeric materials.

I have no criticisms of the paper - there is little to criticize in that this is an "implementation", or "demonstration", of a software suite that the scientific community will find particularly useful. Having said that, because the work is focused on polymers, the authors might want to consider submitting this manuscript to *Macromolecules*, where it might attract more attention and interest. That, of course, is up to the authors and editors to decide. From the point of view of impact, this is a fine paper.

Reviewer #3 (Remarks to the Author):

1. This is an interesting paper describing advanced software for automatic setup and initiation of molecular dynamics simulations of polymers, nanomaterials and biological macromolecules. In particular, it is aimed at facilitating generation of initial models and parameters, thereby enabling high throughput simulations of a wide range of complex systems.

2. This is an important area of computational nanoscience and biophysics. The work described will undoubtedly be of interest to researchers in this area. This does raise an interesting question as to whether, given the increasing importance of molecular simulations in studies of soft materials and complex (bio)molecular systems, a more general awareness of such methodology would be timely. I would argue in favour of raising awareness of molecular simulations in this broad area. However, I realise a decision on whether this study will be of more general interest is perhaps an editorial decision.

3. The Introduction to the paper provides a sound – if somewhat prosaic – account of the need for such an approach. The methods are adequately described.

4. Three key example applications are outlined:

a. Polymer melts: using a combination of atomic level (GROMOS) and coarse-grained (Martini3) simulation. Following initial model generation, energy minimizations for 10 replicas provides an ensemble of configurations, the end-to-end distances of which agree with a wormlike chain model. Some limited MD simulations are run for a subset of systems.

b. Polymeric Li ion battery: this is beyond my direct area of expertise, but seems to be suitably complex.

c. Lipid vesicle with a phase separated interior: it is impressive that this model can be generated. As the authors note this is very challenging with existing methods, and is of biological relevance given current interest in phase-separated membraneless organelles within cells. biological systems. Again the outcome of the modelling is evaluated via equilibration and short MD. (Note – the results are shown in Fig. 5 not as stated Fig. 6).

5. The size and complexity of the systems which can be generated and simulated by this approach is very impressive, and clearly has a wide range of applications in molecular simulations of complex soft materials. The nature of the paper is such that the key example applications are presented i.e. models are generated and shown to provide starting points for plausible simulations. However, 'production runs' with these models, alongside detailed analysis and comparison with experiments are not provided. This is perhaps the main weakness of the paper, reflecting its methodological/software description nature.

All changes related to the review are highlighted in cyan in the main manuscript and a comment is added at the beginning of each such section indicating to which answer that section belongs.

Reviewer #1 (Remarks to the Author):

In this work, the authors who have developed Martini CG models for long time are now developing general schemes to generate input files and realistic starting coordinates for complex bio and nano materials. Since MD simulations and molecular models have become matured now, modeling of complex starting structures in bio and nano materials are bottle necks in their simulations, as the authors say. Their Python suite, polyply, have several unique features, such as automatic generation of parameters for complex polymeric topologies, and a generic multi-scale random walk protocol for building structures. I think that this approach is interesting and useful for many applications in particular, polymer systems.

However, I have several concerns to be clarified before the publication of this work in this journal.

1) Force field independency of polyply

In the manuscript, the authors mention that polyply is applicable to nano and bio-polymers regardless of force field parameters. However, on page 12, they mentioned that parameter generation is limited to Gromacs input files only. For bio- and nano-material simulations, there are many other programs, for instance, AMBER, CHARMM, NAMD, LAMMPS, OPENMM, etc. If the authors indicate that this program is applicable for any force field parameters, they should prepare input file-parsers for other force-field as soon as possible. If this is not so easy, the authors should not emphasize the force-field independency in the manuscript.

Answer 1.1: We thank the reviewer for this comment and acknowledge that there are many other programs for running MD simulations, which are frequently used. However, in terms of force-field independence we would argue that the force-field is (or should be) independent of the simulation program. For example, CHARMM, Amber, OPLS and GROMOS are all force-fields which have their own simulation code, but are also implemented and working in GROMACS. On the other hand, programs like OPENMM in fact parse input files from GROMACS as well as the input files of other programs. In addition, there exist several programs, which facilitate the conversion of input files between the different MD programs. They are written by experts of the file format and work reliable. Thus, every user can convert input files from and to GROMACS format if needed. However, to take the reviewers critique into account we adjusted manuscript to include a reference to a paper, which discusses the challenges of converting input files between different MD engines. In addition, we add Table S2 in the Supplementary Material, which lists programs for the conversion of input files from and to GROMACS format. We hope that this will help users who want to use different software than GROMACS to convert polyply output reliably.

By generating input files for Martini2, Martini3, Gromos56A7, 2016H66, OPLS, and Amber polymers we would consider it fair to emphasize that the program is force-field independent within reasonable limitations. These limitations being that certain highly customized force-fields have non-standard bonded and non-bonded interactions, which our code does not account for in the coordinate generation. Implicitly this is pointed out by writing that we are limited to GROMACS

input. However, the input parameter generation code parses a native input file format of the vermouth library, which in principle can be extended with those customized functions. To do so users can simply open an issue on the vermouth GH page or create an appropriate pull request.

For future development, we consider it more useful to work with users that are interested in those special potentials to implement them in polyply, because these are often not well documented and require expert knowledge to avoid producing bad structures. For example, within the polyply double-stranded DNA extension we are looking into generating input parameters and coordinates for the 3SPN DNA model by de Pablo and coworkers, which utilizes some of those specialized interactions. We added a statement in the discussion clearly pointing out that the coordinate generation is not capable of handling those custom bonded or non-bonded interactions.

2) Applicability for bio-polymers

For this reviewer, it is not so clear which biomacromolecules could be handled using this software. I would like to hear the current status in the program. I would like to hear which biomacromolecules are available using this software. I think that multi-scale random walk protocols for building structure is especially useful for polymer systems, while I wonder how it is applicable to bio-polymers, such as proteins, DNA, and RNA systems. Since MD simulations of cellular environments are one of the new trends, clarifying the applicability of complex biological systems using the software is meaningful for computational biophysicists.

Answer 1.2: We apologize that the manuscript did not clearly enough demonstrate the applicability of polyply to bio-macromolecules in the context of computation biophysics. First, we would like to point out that polyply has already been used for generating bio-macromolecules. We have published a protocol chapter for extending proteins by PEGylated lipids (DOI: 10.1007/978-1-0716-0892-0_18), and used polyply to generate the Glycans covering the SARS-Cov2 Spike protein (DOI: 10.1101/2021.09.15.459697), which has been used in our simulation of the viral envelope of the SARS-Cov2 virus. To take the reviewers critique into account we clearly highlighted both examples in the manuscript. Furthermore, the third test-case, where we setup a system of PEG Dextran inside a vesicle, is a typical model system used in experimental biophysics as model for cellular interior. Just a few weeks ago a new preprint using this system has been published (DOI: 10.21203/rs.3.rs-827501/v1). Therefore, we would argue that this is already a meaningful and useful system for computational biophysics.

However, to further demonstrate that polyply is applicable to bio-macromolecules as well as polymers found in materials science, we have extended our library to include single-stranded DNA parameters within the Martini2 force-field and the Amber compatible Parmbsc1 force-field, as well as parameters for disordered proteins within the Martini3 force-field. Proteins with secondary and tertiary structure, as well, as double stranded DNA (dsDNA) are currently not possible to model with polyply. We now mention this limitation more clearly in the discussion section. However, within the development of Martini3 DNA&RNA parameters we are working on extending polyply to dsDNA. Due to the importance of dsDNA in biological systems as well as the vast amount of data known on dsDNA we consider it more appropriate to keep the implementation separate.

Additionally, we have added a new test-case in the examples section. The test-case highlights that using a new feature to set the persistence length in polyply, ssDNA conformations are generated, which are in good agreement with experimental measurements. Furthermore, we note that overall conformations show reasonable overlap with the conformations visited in an unbiased all-atom simulation (see supplementary material section 4.2). Finally, we have tested ssDNA in a complex test case. We generated input parameters and coordinates for the complete genome of the porcine virus inside the virus capsid at all atom level. The simulation could be run without issues for 60 nanoseconds.

3) Backmapping to build realistic starting structures.

In complex polymer systems, it is not straightforward to obtain good (low-energy) starting structures. Without special cares, starting structures could have many problems. Then, this requires long relaxation times or generates unstable trajectories in the MD simulations. I would like to hear that in their backmapping, how they elaborate to avoid artificial unstable structures. For instance, entanglement of polymers or ring-penetrations are often pointed out as starting structures to be avoided. If the backmapping describing in Figure 2 includes any techniques, I would like to ask the authors extend such descriptions in main text or supporting information.

Answer 1.3: Avoiding high energy structures is a multi-level approach within polyply that is not only related to the backmapping procedure. However, we agree that this warrants more details and an extended discussion. To improve explanation of the workflow behind the structure generation we have introduced a new section in the Supplementary Material namely S3, which provides flowcharts of the algorithms mentioned in the main paper as well as benchmarking of some critical aspects. Below we like to outline the multi-level approach and how the information presented in S3 supports our conclusion that polyply can offer good low energy starting structures.

Level 1: If the random-walk places beads too close to each other there is no way for the back-mapping protocol to fix this and final structures will overlap. This typically leads to crashes in the energy minimization. In general, even though this can happen probabilities are very low within the many test cases we generated when using default settings. This means the volumes obtained from our generic CG model are sufficient to place chains at an adequate distance. However, it can in principle happen especially if residues are very large e.g., in the order of a C60 molecules. In this case the user can lower the rather liberal criterion for the maximum force, which results in larger spacing between residues. Sometimes running the algorithm again will also fix the problem. As polyply is comparatively fast, this is no problem in our opinion. To make this known to users we have established an FAQ section (https://github.com/marrink-lab/polyply_1.0/wiki/FAQs) on the wiki that deals with these kinds of problems and added the link also into the main paper.

Level 2: Polyply backmaps entire residues. These are optimized prior to the backmapping stage and fix several important conformations at the residue level. For example, integrity of rings is introduced at this stage. We have added a flow-chart of the algorithm as well as some benchmarking of the coordinate generation protocol. We find that overall, it produces very good templates in a consistent fashion (see Supplementary Information section S3.1). The better the template of the residue the less artificial conformations within a residue we find in the final molecule.

Level 3: Finally, at the backmapping stage first the rotation of the template is optimized such that atoms which form bonds to another residue are oriented towards that other residue. Secondly, all coordinates are scaled to the center of mass by a factor of 0.45. In the last step the user needs to run an energy minimization which then very efficiently expands the residue outwards avoiding potential overlaps and artifacts such as ring penetrations.

As the reviewer mentioned ring penetrations explicitly, we have analyzed the PS melts that we produced in example 1. As now discussed in Supplementary Material S3.3, the lowest COM distance between two rings is found to be 0.3 nm. This is significantly larger than the 0.2 nm that would be found for interlocked rings. Since the melts contain up to 50,000 ring fragments of which not a single one was found to be interlocked, we conclude that the algorithm is very efficient at preventing this artifact.

4) Spatial geometry of complex initial structures

In the manuscript, they showed three examples. Polymeric lithium ion battery and lipid vesicle with liquid-liquid phase separated interior are very complicated simulation systems. For my understanding, there is a big gap between the explanation in sections 2 and the examples in section 3. To reduce the gap, I suggest the authors to add schemes for modeling the two systems in supporting information, describing the modeling protocols step-by-step. Also, it is helpful to include the help of other software (like TS2CG) for the modeling. It should be clarified what is the input data and what is the output in each step.

Answer 1.4: We thank the reviewer for pointing out this discrepancy. We added the workflow schemes as mentioned in the supplementary material (Figure S7, S8, S10). Regarding the third test-case we took it one step further and turned it into a tutorial. The tutorial is in form of a jupyter notebook and users can download it from our webpage (<http://cgmartini.nl/index.php/2021-martini-online-workshop/tutorials/559-8-polyply>) and simply run it on their local machines. The notebook in detail demonstrates all aspects of how to generate the system. Furthermore, we also added several other tutorials that illustrate how to use polyply to generate the examples mentioned in the paper as well as several others. Overall, we now have 6 tutorials illustrating most aspects of how to use polyply. The tutorials can be found on our GH wiki page (https://github.com/marrink-lab/polyply_1.0/wiki) and are kept up to date.

5) Restraints from experimental information

In biomolecular modeling, inclusion of experimental information as restraint is essential. For instance, chromatin modeling is possible only when one incorporates the distance-restraints obtained from Hi-C experiments. Using polyply, can we introduce such restraint forces originally observed by experiments?

Answer 1.5: The geometrical constraints allow to take experimental information such as phase separation or cellular walls efficiently into account. Even though these are more qualitative examples of experimentally obtained information, they are nevertheless important also for biophysical system. For the future we are working on extending the geometrical constraints to arbitrarily complex surfaces by using triangulated surfaces or voxelized descriptions of a surface. This would allow polyply to consider larger and more complex boundary surfaces such as highly

curved lipid bilayers very efficiently. With that extension it would be possible to for example fill the model of our mitochondria (DOI: 10.1038/s41467-020-16094-y) that was recently created in our group with DNA given the pending DNA implementation of course.

Inspired by the reviewers comments we also implemented a new algorithm, which allows to take specific distance restraints into account in the random walk. Section 3.2 of the supplementary material contains some benchmark results on a generic polymer. Those results show that now the random-walk can meet quite effectively several distance restraints. This feature allows users to take into account data from Hi-C experiments or for example NMR experiments. It also allows to create large macro-cycles such as circular DNA very efficiently. Of course, the larger the number of restraints the less efficient the random-walk will be and there are certainly limitations. But we are looking forward to exploring those limitations in future applications and improve upon the current algorithm.

In summary, I believe that this tools even at the current stage could be regarded as a very useful tool in nano- or polymer sciences. If the authors would like to emphasize the applicability of this software in biological sciences, I would like to ask them to clarify the above points indicating the current functions or future perspectives.

Answer 1.6: We thanks again for the suggestions of the reviewer, which improved the overall quality of the paper and the code. We hope that with the new examples emphasized in text and also exclusively generated during this revision (e.g. virion capsid filled by single strand DNA), we now clearly indicate the applicability of polyply to biological systems.

Reviewer #2 (Remarks to the Author):

This is a really, really useful piece of work, which outlines a general framework and the corresponding workflows for setting up molecular dynamics simulations of polymeric materials. I anticipate the manuscript will be highly cited and widely used by the polymer community.

In terms of content, the actual methods or results are not new. The work focuses instead on describing a "tool" that will enable or facilitate research on polymeric materials.

I have no criticisms of the paper - there is little to criticize in that this is an "implementation", or "demonstration", of a software suite that the scientific community will find particularly useful. Having said that, because the work is focused on polymers, the authors might want to consider submitting this manuscript to *Macromolecules*, where it might attract more attention and interest. That, of course, is up to the authors and editors to decide. From the point of view of impact, this is a fine paper.

Answer 2.1: We thank the reviewer for his overall positive evaluation of the paper. To further illustrate the applicability of the code, we added new examples more focused on biological applications (see answer 1.2 to reviewer 1) and also new features and force-field options to the code (see answers 1.1 and 1.3 to reviewer 1). Considering the range of applications, accuracy, and

computational performance of polyply, we emphasize here our intend to publish this work for more a general and multidisciplinary audience, which is the main target of Nature Communication.

Reviewer #3 (Remarks to the Author):

1. This is an interesting paper describing advanced software for automatic setup and initiation of molecular dynamics simulations of polymers, nanomaterials and biological macromolecules. In particular, it is aimed at facilitating generation of initial models and parameters, thereby enabling high throughput simulations of a wide range of complex systems.
2. This is an important area of computational nanoscience and biophysics. The work described will undoubtedly be of interest to researchers in this area. This does raises an interesting question as to whether, given the increasing importance of molecular simulations in studies of soft materials and complex (bio)molecular systems, a more general awareness of such methodology would be timely. I would argue in favour of raising awareness of molecular simulations in this broad area. However, I realise a decision on whether this study will be of more general interest is perhaps an editorial decision.

Answer 3.1: We thank the reviewer for his positive view about our work. Indeed, with the advance of computational methodologies, we also think our work serves as a way to present how capable a tool such as polyply can be to generate complex systems. For sure, this will be a trend in the modeling field applied to different areas, from material science and nanotechnology to biophysicists and medicine. We hope that our current version of the manuscript, which was now expanded in relation to the first version (see answers for reviewer 1), fully convince you and the editor to present polyply to the broad audience of Nature Communications.

3. The Introduction to the paper provides a sound – if somewhat prosaic – account of the need for such an approach. The methods are adequately described.

4. Three key example applications are outlined:

- a. Polymer melts: using a combination of atomic level (GROMOS) and coarse-grained (Martini3) simulation. Following initial model generation, energy minimizations for 10 replicas provides an ensemble of configurations, the end-to-end distances of which agree with a wormlike chain model. Some limited MD simulations are run for a subset of systems.
- b. Polymeric Li ion battery: this is beyond my direct area of expertise, but seems to be suitably complex.
- c. Lipid vesicle with a phase separated interior: it is impressive that this model can be generated. As the authors note this is very challenging with existing methods, and is of biological relevance given current interest in phase-separated membraneless organelles within cells. biological systems. Again the outcome of the modelling is evaluated via equilibration and short MD. (Note – the results are shown in Fig. 5 not as stated Fig. 6).

Answer 3.2: We thanks the reviewer for pointing out the typo. The manuscript text now properly indicates the correct Figure.

5. The size and complexity of the systems which can be generated and simulated by this approach is very impressive, and clearly has a wide range of applications in molecular simulations of complex soft materials. The nature of the paper is such that the key example applications are presented i.e. models are generated and shown to provide starting points for plausible simulations. However, ‘production runs’ with these models, alongside detailed analysis and comparison with experiments are not provided. This is perhaps the main weakness of the paper, reflecting its methodological/software description nature.

Answer 3.3: We thank the reviewer once more for the good feedback about our work. Regarding the weakness mentioned: as the main aim of polyply is to generate good quality initial configuration for MD simulations, the paper was not focused on performing extensive sampling and then studying the systems described. This is out of the scope of the paper, but it should be explored in future works dedicated to the applications and not to the computational tool. So, we try to emphasize here polyply and the accuracy of the initial configurations generated. For instance, the new test-case with ssDNA highlights that polyply conformations are generated in good agreement with persistence length of experimental measurements. Furthermore, we show that the persistence length of the initial configurations can also be adjusted to match specific force-fields (see answer 1.2 to reviewer 1 and Supplementary Material S4.2).

REVIEWERS' COMMENTS

Reviewer #1 (Remarks to the Author):

I believe that Polyply is a big step, which makes simulations of more complicated systems easily available. Their responses to the three reviewers' comments and criticisms are satisfactory and the revised manuscript has become more clear about what Polyply can do for the modeling of complicated nano- and bio-molecules. I really hope the authors to continue the development and improvement of Polyply for long time as they have done for Martini models.